# Which clinical research questions are the most important? Development and preliminary validation of the Australia & New Zealand Musculoskeletal (ANZMUSC) Clinical Trials Network Research Question Importance Tool (ANZMUSC-RQIT)

William J. Taylor[1,2,3]*, Robin Willink[1], Denise A. O'Connor[4,5], Vinay Patel[1], Allison Bourne[4,5], Ian A. Harris[6,7,8], Samuel L. Whittle[4,9], Bethan Richards[7,8], Ornella Clavisi[10], Sally Green[11], Rana S. Hinman[12], Chris G. Maher[13,14], Ainslie Cahill[15], Annie McPherson[16], Charlotte Hewson[16], Suzie E. May[16], Bruce Walker[17], Philip C. Robinson[18†], Davina Ghersi[19], Jane Fitzpatrick[20], Tania Winzenberg[21], Kieran Fallon[22], Paul Glasziou[23], Laurent Billot[24], Rachelle Buchbinder[4,5]

1 University of Otago, Wellington, New Zealand, 2 Hutt Valley District Health Board, Lower Hutt, New Zealand, 3 Hauora Tairawhiti, Gisborne, New Zealand, 4 Department of Epidemiology and Preventive Medicine, School of Public Health and Preventive Medicine, Monash University, Victoria, Australia, 5 Monash-Cabrini Department of Musculoskeletal Health and Clinical Epidemiology, Cabrini Health, Victoria, Australia, 6 Whitlam Orthopaedic Research Centre, Ingham Institute for Applied Medical Research, Liverpool, Australia, 7 School of Clinical Medicine, UNSW Sydney, Liverpool, NSW, Australia, 8 Institute of Rheumatology and Orthopaedics, Royal Prince Alfred Hospital, Sydney, NSW, Australia, 9 The Queen Elizabeth Hospital, Adelaide, SA, Australia, 10 Musculoskeletal Australia, Melbourne, VIC, Australia, 11 School of Public Health and Preventive Medicine, Monash University, Melbourne, VIC, Australia, 12 Centre for Health, Exercise and Sports Medicine, Department of Physiotherapy, School of Health Sciences, Faculty of Medicine Dentistry & Health Sciences, The University of Melbourne, Melbourne, VIC, Australia, 13 Sydney School of Public Health, University of Sydney, Sydney, NSW, Australia, 14 Institute for Musculoskeletal Health, Sydney, NSW, Australia, 15 Arthritis Australia, Sydney, NSW, Australia, 16 Consumer Partner, ANZMUSC, Melbourne, Australia, 17 Emeritus Professor in the College of Science, Health, Engineering and Education (SHEE), Murdoch University, Murdoch, WA, Australia, 18 University of Queensland School of Medicine, Brisbane, QLD, Australia, 19 National Health and Medical Research Council of Australia, Canberra, ACT, Australia, 20 University of Melbourne, Melbourne, VIC, Australia, 21 Menzies Institute for Medical Research, University of Tasmania, Hobart, TAS, Australia, 22 ANU College of Health and Medicine, Australian National University, Garran, ACT, Australia, 23 Institute for Evidence-Based Healthcare, Bond University, Gold Coast, Australia, 24 The George Institute for Global Health, Faculty of Medicine and Health, UNSW Sydney, Sydney, NSW, Australia

† Deceased.
* william.taylor@otago.ac.nz

## Abstract

### Background and aims

High quality clinical research that addresses important questions requires significant resources. In resource-constrained environments, projects will therefore need to be prioritized. The Australia and New Zealand Musculoskeletal (ANZMUSC) Clinical Trials Network aimed to develop a stakeholder-based, transparent, easily implementable tool that provides

**Data Availability Statement:** Data files for the best-worst scaling survey and the validation studies are publicly available from 10.6084/m9.figshare.19545658.

**Funding:** This study was funded by the Australian National Health and Medical Research Council (NHMRC) Australia & New Zealand Musculoskeletal (ANZMUSC) Clinical Trials Network Centre of Research Excellence (APP1134856). RB is supported by an NHMRC Investigator Fellowship (APP1194483). RSH is supported by an NHMRC Senior Research Fellowship (APP1154217). The funders had no role in study design, data collection and analysis, decision to publish, or preparation of the manuscript.

**Competing interests:** The authors have declared that no competing interests exist.

a score for the 'importance' of a research question which could be used to rank research projects in order of importance.

## Methods

Using a mixed-methods, multi-stage approach that included a Delphi survey, consensus workshop, inter-rater reliability testing, validity testing and calibration using a discrete-choice methodology, the Research Question Importance Tool (ANZMUSC-RQIT) was developed. The tool incorporated broad stakeholder opinion, including consumers, at each stage and is designed for scoring by committee consensus.

## Results

The ANZMUSC-RQIT tool consists of 5 dimensions (compared to 6 dimensions for an earlier version of RQIT): (1) extent of stakeholder consensus, (2) social burden of health condition, (3) patient burden of health condition, (4) anticipated effectiveness of proposed intervention, and (5) extent to which health equity is addressed by the research. Each dimension is assessed by defining ordered levels of a relevant attribute and by assigning a score to each level. The scores for the dimensions are then summed to obtain an overall ANZMUSC-RQIT score, which represents the importance of the research question. The result is a score on an interval scale with an arbitrary unit, ranging from 0 (minimal importance) to 1000. The ANZMUSC-RQIT dimensions can be reliably ordered by committee consensus (ICC 0.73–0.93) and the overall score is positively associated with citation count (standardised regression coefficient 0.33, p<0.001) and journal impact factor group (OR 6.78, 95% CI 3.17 to 14.50 for 3rd tertile compared to 1st tertile of ANZMUSC-RQIT scores) for 200 published musculoskeletal clinical trials.

## Conclusion

We propose that the ANZMUSC-RQIT is a useful tool for prioritising the importance of a research question.

## Introduction

Arthritis and other musculoskeletal (MSK) conditions account for the greatest proportion of years lived with disability (17.1%) of all fatal and non-fatal diseases globally [1]. They affect 24% (~6.1 million) of Australians [2]. There is a large mismatch between MSK disease burden and measures to address it, including a failure to effectively translate the results of research into practice, and a paucity of implementation trials [3, 4].

Compared with other health priorities there is also relatively little research funding and capacity. For example, in Australia, only 1% of the >$300m of Medical Research Future Fund (MRFF) funding for studies with a disease-specific focus was awarded to MSK conditions from 2016 to Sept 2019 [5], while the National Health and Medical Research Council (NHMRC) invests between 3 to 10 times less in arthritis and musculoskeletal research than in research relating to other high burden conditions like cancer, cardiovascular disease, mental health and diabetes [6]. Similarly, in the United States, funding for the National Institute of

Arthritis and Musculoskeletal and Skin Diseases (NIAMS) has never been more than 2% of the National Institutes of Health annual disbursement [7].

Furthermore, similar to other healthcare areas [8], much research effort in the MSK field is wasted [9]. The reasons are multifactorial and include insufficient consideration of questions based upon the greatest burden of disease, failure to take account of existing evidence, methodological shortcomings in the conduct of the research, publication bias, biased reporting, and failure to effectively translate the results of research into practice [8].

The Australia and New Zealand Musculoskeletal (ANZMUSC) Clinical Trials Network was formed in 2015 to address this deficit in MSK research and evidence [4]. ANZMUSC members include researchers, clinicians from various disciplines and consumers with MSK health conditions. Key aims of ANZMUSC are to help identify important research questions relevant to MSK health, provide a platform to connect MSK researchers, and reduce research waste and to facilitate a focus of resources on the most important MSK research.

To better understand current methods used in priority setting for MSK research, members of ANZMUSC first undertook a scoping review [10]. This showed that methodological limitations and lack of actionable research questions limited the usefulness of existing approaches. To improve upon previous efforts, a structured, transparent process to rank the levels of merit of potential research questions was required. Such a process should ideally embed diverse stakeholder perspectives, be easy to implement and be current. This process could be operationalized as a multi-attribute scoring tool, underpinned by stakeholder preferences at each step of development. Therefore, we now describe the development and preliminary validation of such a tool–the Research Question Importance Tool (RQIT). This tool is intended to be used by clinical trial networks or research proposal evaluation committees to assess the importance of the question being posed by the research proposal. It could also be used by clinical researchers to understand the key factors that underpin the importance of their research question.

## Methods

This multi-stage mixed-methods study encompassed six linked projects (Fig 1): (1) a modified Delphi study to identify dimensions and attributes of highly important research questions, (2) a consensus workshop to develop a preliminary framework of dimensions and levels for these dimensions, (3) an inter-rater reliability study to confirm that these levels can be ordered

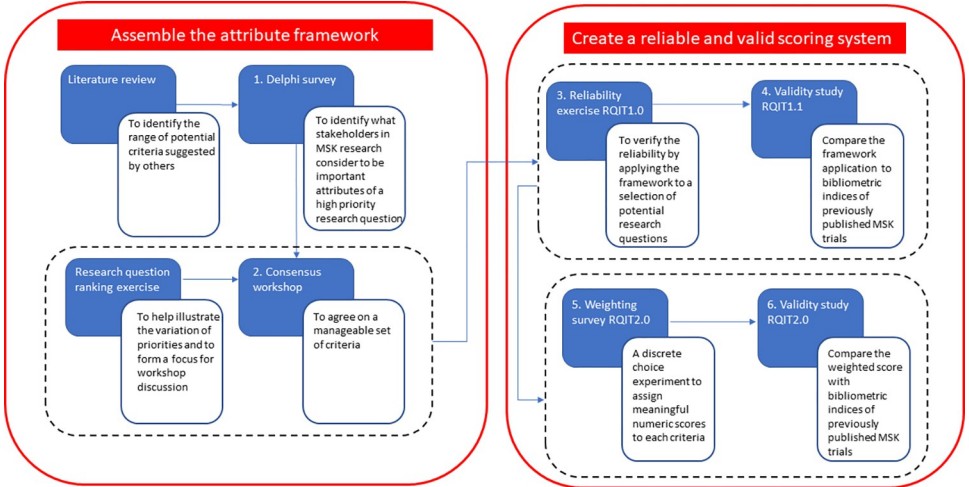

**Fig 1. A schematic of the overall study structure.**

reliably, (4) a test of the validity of dimension levels, (5) a weighting of the dimension levels using discrete choice methods to form individual measures of importance, and (6) a final validation study using summed utilities of the tool compared to places of publication (journal impact factor) and citation counts.

The development of the tool was iterative, allowing for changes to the tool as the study proceeded. So, for example, one dimension of the tool was removed after project #4 to form an evolved version of the tool (ANZMUSC-RQIT) for projects #5 and #6. Consumers (health-service users) or consumer advocates were involved at every stage of the project, including generation of possible attributes of research importance (9% of participants in the Delphi survey), consensus development of the initial RQIT framework (18% of participants in the consensus workshop), weighting of specific elements of the framework using a novel best-work scaling method (12% of participants in the discrete choice survey) and final authorship of this manuscript (20% of authors). Consumers were identified from membership of ANZMUSC and all had lived experience of musculoskeletal disorders.

## 1. A web-based modified Delphi study to generate and reach agreement on a set of attributes that stakeholders believe make a research question important

An iterative, repeated web-based survey using REDCap software was conducted among ANZMUSC members (n = 106, November 2016). The first survey asked participants to generate up to 10 characteristics of research question importance using the prompt:

*When thinking about research, what do you believe makes a research question really important? This question is not about specific examples of research questions that you think are important, but why you think the question is important. Please list as many as you think are appropriate.*

The raw text of all responses was organised into items of similar meaning and duplicates were removed. The wording of some responses was changed to improve interpretation. This process was initially undertaken by a single researcher and then checked by three other researchers in an iterative way until disagreements were resolved.

In the second round of the survey, participants were provided with the final list of determinants identified in round one and asked to indicate their agreement on a numerical rating scale ranging from 1 (strongly disagree) through 9 (strongly agree) to the question:

*Please indicate how strongly you agree or disagree that the item reflects the IMPORTANCE of a research question.*

In the third and fourth rounds of the survey, items were selected for re-rating if the median rating was 4 through 6 (indicating uncertain opinion) or if there was significant disagreement among respondents, as defined by the RAND/UCLA disagreement index >1. This index is derived from the interpercentile range (30th to 70th percentiles) of ratings, adjusted for the symmetry of responses and compared to a value obtained from empirical observation of panel consensus exercises [11].

## 2. A face-to-face workshop to develop the preliminary attribute framework (RQIT) and descriptors of categories for each attribute

A 2-day workshop was held in Brisbane (November 2017) and facilitated by DAO and WJT. Participants were invited based on their interest in designing, funding or participating in MSK

clinical trials. Selection aimed to ensure a breadth of professional disciplines and backgrounds, including consumers, funders, clinicians and researchers. The workshop consisted of (a) pre-workshop data-gathering exercises (the published scoping literature review, the Delphi survey of ANZMUSC members, and a research question ranking exercise by workshop participants); (b) review of these pre-workshop activities; (c) facilitated discussion on what makes a research question important using selected research question examples from the ranking exercise; (d) small group activities that developed key dimensions and categories of each dimension that described the underlying basis of research question importance; (e) facilitated discussion and voting for alternate descriptions of the dimensions and categories. RQIT was the main outcome of this process.

The pre-workshop research question ranking exercise was designed to facilitate participant thinking about what made a research question important. From 227 research questions obtained from the systematic literature review [10], suggestions from ANZMUSC members and studies presented or endorsed by ANZMUSC, 30 research questions were selected to encompass a range of research importance (graded subjectively as 1 = not important through 5 = important). Each workshop participant was asked to rank these research questions in order of importance, prior to the workshop event.

## 3. Rater reliability study to determine whether research proposals could be reliability categorised using the preliminary attribute framework (RQIT)

This study involved 32 ANZMUSC members evaluating 14 hypothetical research questions with accompanying information as might be found in the justification section of a real research proposal document (see S2 Appendix). For each research question, raters were required to categorise the research question according to the six domains. The six domains were labelled: stakeholder importance, patient burden, society burden, intervention effect, scalability and uptake, and equity.

Using an ICC framework (2-way random-effects, absolute agreement) to assess inter-rate agreement, it was determined that 30 raters and 14 research scenarios would be required to estimate an ICC of 0.75 with sufficient precision (95% confidence interval width of 0.3) [12].

An ICC is reported, both for individual scores and for average scores. In addition, since the categorisation of each attribute would normally be done by committee consensus, rather than by individual members, 5 'committees' were randomly selected from the 26 individual respondents (4 groups of 5 and 1 group of 6) and the average rating of each 'committee' was used to calculate the ICC (single measure scores only are reported). Only respondents with complete data for every research question scenario were included in this calculation (n = 26).

Gwet's AC2 is also reported as an agreement index for categories amongst multiple raters [13]. It can be interpreted in a similar way to Cohen's kappa but is less prone to kappa paradoxes such as low values when agreement is high and is less affected by event distribution [14].

## 4. Testing the validity of the RQIT framework by comparing it to the citation count and impact factor of journals in which MSK research has been previously published

A test of the construct validity of the RQIT framework was conceptually challenging. It was difficult to identify a meaningful and measurable index of 'research question importance' to compare against RQIT. One potential approach is to make the assumption that research that asks more important questions will have greater 'impact'. However, there is no broadly accepted standard for how to measure research impact [15]. Although there are well-known limitations to the use of bibliographic indices to measure research impact, particularly with

regards to the broader concepts of actual health gain or clinical practice, it was thought to be too difficult to undertake a broad and multidimensional assessment of research impact for the limited purpose of assessing construct validity of RQIT. Therefore, we limited the measurement of impact to journal impact factor and citation count, making the reasonable assumptions that more important research questions will likely be answered in higher impact factor journals and will likely have been more frequently cited. An association between the bibliometric indices and RQIT could be seen as a minimal requirement for construct validity.

Wording and categories of the tool were adjusted following the first reliability study. The 37 journals selected for this study were from the same list as the scoping review of MSK trials identified by Bourne et al. [10] and were ordered by the 2015 ISI Web of Science Impact Factor (IF). The top 5 ranking journals and bottom 5 ranking journals were selected:

- Top 5 (High Impact Group): New England Journal of Medicine, Lancet, Journal of the American Medical Association, British Medical Journal, Annals of Internal Medicine

- Bottom 5 (Low Impact Group): BMC Musculoskeletal Disorders, Journal of Bone & Mineral Metabolism, European Spine Journal, Archives of Physical Medicine and Rehabilitation, Clinical Rehabilitation

A literature search to obtain randomised controlled trials relating to MSK health from each journal via Ovid Medline was performed. The search was performed with publication dates between 2014 and 2018 using the terms "randomized" and "trials". These filters were linked with search terms including: "arthritis," "gout," "juvenile idiopathic arthritis," "osteoarthritis," "rheumatoid arthritis," "spondyloarthritis," "fracture," "tear," "pain," "lupus," "sciatica," "osteoporosis," "systemic sclerosis," "vasculitis" and "fall." An additional search using the same filters and search terms was performed on each journal's website.

The list of trials was screened to only include articles on randomised controlled trials, published between 2014 through 2018 (5 years) that related to MSK health. The lists from low impact journals and high impact journals were separately ordered chronologically and we selected the first 100 articles in each list starting from the beginning of 2014.

Each article was evaluated using the RQIT and the Cochrane Risk of Bias Tool (RoB2.0) [16], as a check to confirm that trials with lower potential risk of bias were published in higher impact journals. For analysis of place of publication (high or low journal impact factor), a univariate analysis using Chi-square (or Fisher's Exact Test) and Cramer's V statistics were undertaken for each dimension of each of the two tools. Cramer's V is calculated as the square root of chi-square divided by sample size, times the smaller of (rows—1) or (columns—1) [17]. It is interpreted in the same way as a correlation coefficient. For each dimension of each tool, the null hypothesis for each statistical test was that there was no difference in the frequency distribution of each dimension-level for articles published in the high journal impact factor group compared to the low impact factor group.

The unweighted overall RQIT score was calculated as the simple sum of the levels across the 6 dimensions (all dimension levels scaled to range 0 to 3) giving a possible score range 0 to 18. Logistic regression was used to test the hypothesis that there was no association between the overall RQIT summed score and high and low impact journal groups.

The citation count (to January 2020) for each article was obtained from Clarivate Web of Science and compared to RQIT dimension levels using boxplots and ANOVA (citation counts were natural log transformed because of skewed distributions). The overall summed RQIT score was compared to citation count using linear regression.

The combined results led to ANZMUSC-RQIT as described in the results section.

## 5. Estimation of the utilities (scores) for each type of research question in ANZMUSC-RQIT

ANZMUSC members (n = 381, April 2021) were invited to take part in a novel best-worst scaling preference survey. Participants undertook a number of choice-tasks online, each of which involved considering three options simultaneously and choosing the options they thought represented the 'most important' and 'least important' research questions. Each option consisted of a pair of dimension-level combinations called 'elements'. For example, the pair of elements (A3, B1) represents a research question with level 3 of dimension A and level 1 of dimension B. An option is part of a full profile such as (A3, B1, C2, D2, E4) which represents the elements for all five dimensions and which is deemed to have an objective level of merit, called a '(profile) utility'. The overall goal was to obtain the estimates of the utilities of all the feasible profiles.

A profile utility was regarded as the sum of utilities of its five elements. Envisaging the existence of objective (element) utilities meant that we could employ standard statistical principles of estimation and could invoke the usual concepts of fixed 'parameters' and variable 'measurement error' when estimating these utilities. The same two dimensions were represented in the three options in any choice-task, which meant that legitimate comparison was possible. For example, the utilities associated with options (A3, B1), (A2, B2) and (A1, B3) were comparable, because the levels of the other dimensions were held constant and were assumed to play no role.

Given the possible combinations of element pairs imposed by the structure of ANZMUSC-RQIT, it was determined that 86 specific choice-tasks were necessary to provide a full factorial design. To reduce respondent burden, participants were randomised into one of three groups, each answering 30 choice-tasks (a random selection of 28 choice-tasks plus two others that everyone answered).

Each response to a single choice-task led to data points from three implied paired comparisons. These were fitted to Thurstone's model of paired comparisons [18] using a bivariate-normal model of measurement error to estimate the element utilities, and thereby to estimate the profile utilities. Two models were fitted. Model 1 'the full model' allowed different people to have different standard deviations of the random fluctuations in their judgements (different standard errors of measurement, SEM). Model 2, the 'simpler model', constrained the SEMs of the respondents to be the same. This model was expressed in different but equivalent ways, Model 2a and 2b, the difference being that Model 2b facilitated the calculation of standard errors by constraining the within-dimension sum of utilities to 0. More details of the models and fitting procedure are described in the S1 Appendix [19–23].

To make the scores whole numbers while retaining precision, convenience and user familiarity, the utility values were re-scaled using a linear transformation to range from 0 (lowest possible scoring profile) to 1000 (highest possible scoring profile).

## 6. Repeating the validity test using the profile (summed across dimensions) utilities

The ANZMUSC-RQIT scores for the same 200 articles selected previously from 5 high impact factor journals and 5 low impact factor journals (step #4) were recalculated using the weighted scores (utilities) derived in step #5. The validity of the ANZMUSC-RQIT was assessed by the extent of association between the profile score for each article and whether it was published in a high or low impact factor (IF) journal, using logistic regression. ANZMUSC-RQIT scores were divided into tertiles given the arbitrary scale unit of the tool and journal IF grouping was the dependent variable. A similar analysis was performed for the original (unweighted scoring)

of RQIT. In addition, the ANZMUSC-RQIT profile scores were compared to the citation counts arising from each article using linear regression. Citation counts were natural logarithm transformed because of the markedly skewed distribution.

## Results

### Delphi survey

The first survey was completed by 66 ANZMUSC members and elicited 347 possible determinants of research question importance which were organised into 43 non-duplicate items for subsequent rating. By the 3rd Delphi iteration (4[th] survey), there were 54 participants (clinician 22%, researcher 57%, consumer 9%, other 11%). The final list of 32 items with median ratings of at least 7 were classed into 5 groups and ordered by median rating (Table 1). These themes contributed 28.8% of the sum of median ratings (Potential for Impact), 24.5% (Broad Appeal), 21.8% (Nature of the Intervention), 15.7% (Population Need) and 9.2% (Project can Deliver).

### Framework development workshop

Workshop participants included the facilitator (DAO), workshop leader (WJT) and 22 participants: 4 consumers or consumer advocates, 3 research funders or insurance providers, 2 sport and exercise medicine physicians, 4 physiotherapist or chiropractic researchers, 4 rheumatologist researchers, 1 orthopaedic surgeon researcher, 2 biostatistician or non-clinical MSK researchers, and 2 primary-care doctor researchers. There was marked variation in how workshop participants ranked the importance of 30 research questions relevant to MSK clinical

**Table 1. Final list of items (sum of ratings) with median rating 7 through 9, by theme from the modified Delphi exercise.**

| Population need (36) | Nature of intervention (50) | Potential for impact (66) | Broad appeal (56) | Project is able to deliver (21) |
|---|---|---|---|---|
| Condition has a high patient and societal burden (8) | Extent to which the intervention could prevent disease or disease progression (8) | Addresses large practice-evidence gap (8) | Agreed importance for funders, consumers and providers (7) | The extent to which the question can be robustly answered (testable) (7) |
| Condition has few effective treatments (7) | Intervention is easily and widely implementable (7) | Results have potential for fundamental shifts in understanding (8) | Extent to which the question is important to consumers (7) | Study is highly likely to be completed (7) |
| Highly prevalent condition (7) | Tests interventions that are easily and widely accessible to consumers (7) | Results likely to lead to real world changes in clinical care (8) | Results are likely to influence government policy (7) | Study will lead to a definitive answer to the question (7) |
| Area of research with little prior work or where prior work is not definitive (7) | Tests intervention in clinical use that have questionable or unknown benefit (7) | Results likely to lead to cost-savings for consumers and/or the healthcare system (7) | Results likely to advance knowledge in other fields (7) | |
| Condition is costly (7) | Addresses the safety of an intervention (7) | Tests interventions with likelihood of significant benefit (7) | Extent to which the question is important to clinicians, consumers and funders (7) | |
| | Test of interventions as they are delivered in real clinical practice (7) | Results have potential for the cure of a health condition (7) | Extent to which the question is important to consumers and clinicians (7) | |
| | Addresses timing of delivery and best combinations of interventions (7) | Study is able to identify the most responsive subgroups (7) | Question is a priority for policy-makers and funders (7) | |
| | | Results likely to improve treatment access, especially equity of access (7) | Extent to which the question is important to clinicians (7) | |
| | | Advances methods for improving implementation into practice and adherence (7) | | |

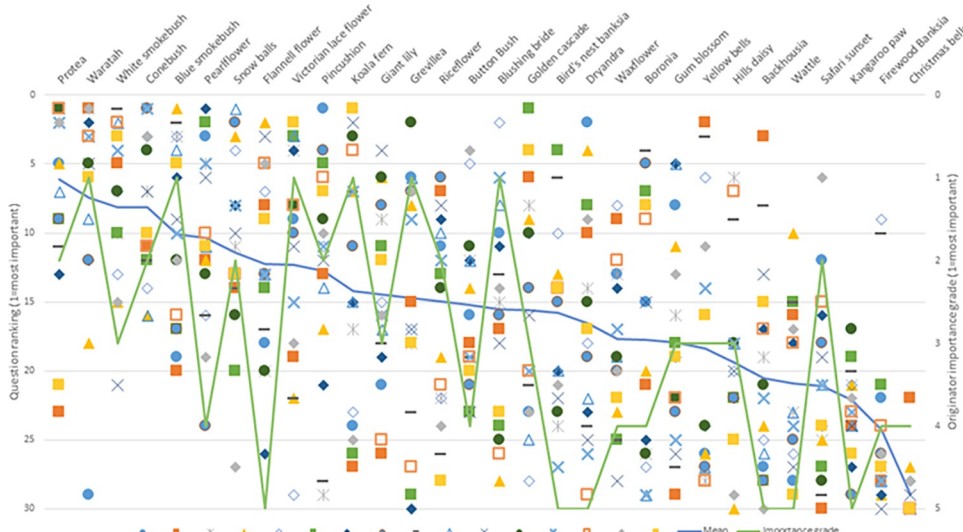

**Fig 2. Distribution of rankings and importance grade of 30 research questions relevant to MSK clinical trials.**
Each research question is labelled as an Australian wildflower and are displayed in order of most highly ranked to least
highly ranked. The blue line shows the mean ranking (left axis) and the green line shows the importance grade
nominated by the source of the research question (right axis). Each data symbol represents a workshop participant.

trials (Fig 2). In discussions, more highly ranked questions tended to reflect high patient burden of disease and prevalence, identification of patients most likely to benefit from an intervention, and widely used interventions that had a poor evidence base. Lower ranked questions tended to reflect animal model studies, testing interventions already known to be ineffective or effective, testing interventions with little prospect of scalability or uptake, or diagnostic research not linked to patient outcomes or benefit.

There was some discussion about the feasibility and design of the study that would address the research question and whether this should come into the evaluation of intrinsic importance of the question itself. It was concluded that these issues should be evaluated separately and are different from the focus on research question importance.

Following small group and plenary discussion, six preliminary dimensions of research question importance were formulated: (A) extent to which the question is important to patients and other health decision-makers; (B) that it addresses an area of high patient burden; (C) that it addresses an area of high social burden; (D) the potential reduction in patient and/ or social burden due to (clinical or implementation) intervention; (E) the potential scalability and uptake of intervention; and (F) the extent to which the question addresses health equity. The RQIT framework is shown in Table 2.

## Inter-rater reliability evaluation

The reliability exercise was started by 34 participants but two did not complete any questions and six did not answer every question. Participants were clinicians (4, 12.5%), clinician-researchers (13, 40.6%), researchers (12, 37.5%) and consumers/consumer advocates (3, 9.4%).

The mean percentage agreement for each dimension (based on the maximum category frequency for each research scenario by dimension) was 66 (*stakeholder importance*), 77 (*patient burden*), 61 (*society burden*), 55 (*intervention effect*), 49 (*implementability*), and 65 (*equity*). As shown in Table 3, the inter-rater reliability of individuals to select categories for each dimension was inadequate (apart from the *social burden* dimension) but the reliability of the average

**Table 2. Research Question Importance Tool (RQIT).**

| Dimension | Category | Descriptor/definition of the category |
|---|---|---|
| A. Extent to which the question is important to patients and other health decision-makers | A1 Not shown to be important to either patients or decision-makers | No clinician or relevant healthcare consumer consultation |
| | A2 Shown to be important to health decision-makers but not patients | Relevant healthcare consumers not consulted or do not rate research question highly |
| | A3 Shown to be important to patients but not decision-makers | Relevant healthcare consumers rate research question highly but other health decision makers do not |
| | A4 Shown to be important to both patients and decision-makers | Relevant healthcare consumers and other health decision makers (not researchers only) rate the research question highly. |
| B. That it addresses an area of high patient burden | B1 Low patient burden | Mild symptoms and little or no associated disability |
| | B2 Medium patient burden | Moderate symptoms and some disability |
| | B3 High patient burden | Significantly disabling, associated with mortality risk or no effective treatments available |
| C. That it addresses an area of high social burden | C1 Condition is rare | (<0.1% prevalence) |
| | C2 Condition is somewhat common | (0.1–1% prevalence) |
| | C3 Condition is common | (1–10% prevalence) |
| | C4 Condition is highly prevalent | (>10% prevalence) |
| D. Potential reduction in patient and/or social burden due to (clinical or implementation) intervention | D1 Symptomatic treatment only and small potential effect size | Intervention has potential to only improve patient symptoms to a modest degree (anticipated effect size <1) |
| | D2 Symptomatic treatment and large potential effect size | Intervention has potential to only improve patient symptoms to a substantial degree (anticipated effect size >1) |
| | D3 Potential for intervention to treat both symptoms and underlying disease pathology | There is a plausible case that some pathophysiological consequences of disease (eg anatomical damage) could be prevented |
| | D4 Potential for cure or fundamental alteration of disease course | There is a plausible case that the disease could be rendered entirely non-active with minimal risk of recurrence, with or without ongoing treatment |
| E. Potential scalability and uptake of intervention | E1 Low potential for scalability and uptake | Prohibitive costs to patient or healthcare system, major systems restructure; and substantial behaviour/belief change by clinicians or patients |
| | E2 High potential for uptake but low scalability | Minimal behaviour/belief change by clinicians or patients required but prohibitive costs to healthcare system, major systems restructure |
| | E3 High potential for scalability but low potential for uptake | Immediately feasible with minimal changes required to healthcare system but requires a substantial change in patients/clinicians beliefs or behaviour or has high direct patient costs |
| | E4 High potential for both scalability and uptake | Immediately feasible and minimal behaviour/belief change by clinicians or patients required |
| F. Extent to which the question addresses health equity | F1 No information | No attempt to address health equity |
| | F2 Not relevant | Discussed using Progress Plus items (O'Neill et al., 2014) but intervention not relevant or appropriate |
| | F3 Somewhat (may have some application to reduce health disparity) | Intervention shown to have some potential application to improving health equity issues |
| | F4 Reducing health disparity is the focus | The intervention is explicitly designed to improve health equity issues |

selection by randomly selected 'committees' of raters was much more acceptable (apart from the *implementability* dimension).

As a result of the reliability exercise, some wording changes and amendments to the number of levels were made to the framework.

## Testing the validity of the RQIT framework

Most RoB2.0 dimensions were associated with journal impact group (S1 Table), especially overall RoB2.0 (Cramer's V 0.41, p<0.001), confirming that studies with lower risk of bias are published in higher impact factor journals.

Table 3. Agreement coefficient and ICC for each attribute (2-way random-effects model, absolute agreement).

| Attribute | ICC (95% CI) for single measure (n = 31 raters with at least 1 rating) | ICC (95% CI) for average measure (n = 26 raters with complete data) | Gwet's AC (95% CI) | ICC for single measure based on 'committees' of raters* |
|---|---|---|---|---|
| Stakeholder agreement | 0.40 (0.24 to 0.64) | 0.94 (0.89 to 0.98) | 0.38 (0.22 to 0.55) | 0.82 |
| Patient burden | 0.42 (0.26 to 0.66) | 0.95 (0.90 to 0.98) | 0.55 (0.41 to 0.68) | 0.77 |
| Society burden | 0.67 (0.51 to 0.84) | 0.98 (0.96 to 0.99) | 0.34 (0.20 to 0.48) | 0.93 |
| Intervention effect | 0.32 (0.18 to 0.56) | 0.92 (0.85 to 0.97) | 0.24 (0.13 to 0.35) | 0.73 |
| Scale and uptake | 0.14 (0.06 to 0.32) | 0.80 (0.63 to 0.93) | 0.14 (0.06 to 0.23) | 0.50 |
| Equity | 0.41 (0.26 to 0.65) | 0.95 (0.90 to 0.98) | 0.37 (0.23 to 0.51) | 0.79 |

* 5 groups of raters were randomly selected (5 in 4 groups and 6 in 1 group) and the average rating for each group was used to calculate the ICC, as if there were 5 raters. The 95% CI is not calculated.

The following RQIT dimensions were also associated with articles published in higher impact factor journals (Fig 3): *patient burden* (Cramer's V 0.29, p<0.001), and *intervention effect* (Cramer's V 0.41, p<0.001). *Social burden* (Cramer's V 0.29, p<0.001) but *implementability* (Cramer's V 0.27, p<0.001) showed a (reversed) association with articles published in lower impact factor journals. The domains of *stakeholders* (Cramer's V 0.16, p = 0.09) and *equity* (Cramer's V 0.16, p = 0.18) were not significantly associated with impact factor group.

## Weighting the choices: Assigning utility estimates in the ANZMUSC-RQIT framework

In light of the poor reliability for assessing the *implementability* dimension and the reversed association with journal impact factor group, it was decided to remove this dimension for the

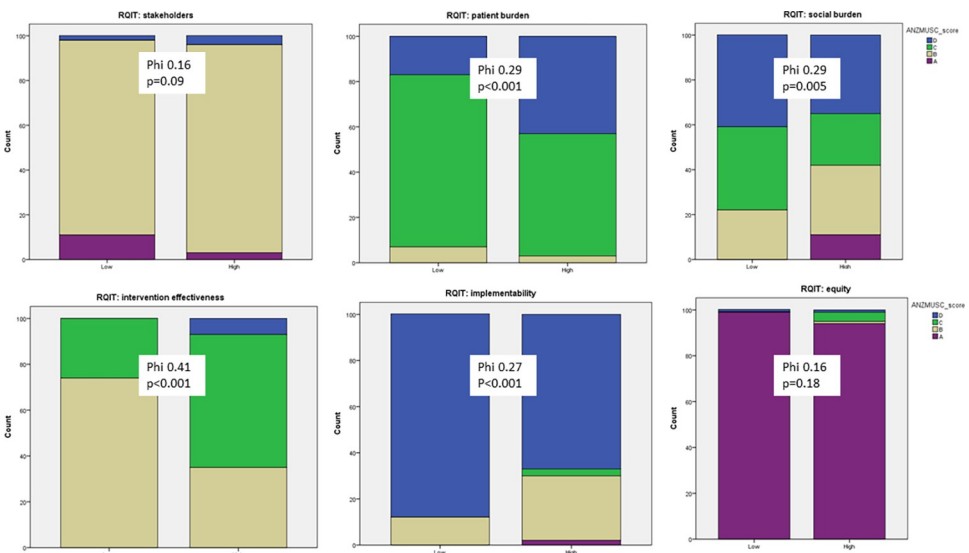

**Fig 3. The level of association between dimensions of RQIT shown as counts of publications by journal impact factor (low, high) and by categories of the RQIT assessment.** The Phi (also known as Cramer's V) statistic can be interpreted in a similar way as a correlation coefficient.

next version of the tool, ANZMUSC-RQIT. This version was used in the modified best-worst discrete-choice survey, as described in subsection 5 of Methods.

Usable responses were received from 41 participants, providing answers to 988 person-choice-tasks. Of these choice-tasks, 982 were answered with both the most important and least important responses, providing three comparisons, while six were answered using only one of these responses, providing two comparisons. There were a total of 2958 comparisons, with each of the 86 choice-tasks being answered between 27 and 102 times. Participants were clinicians (2/41, 5%), clinician-researchers (15/41, 37%), researchers (18/41, 44%), consumers (5/41, 12%) and one (2%) health policy manager.

The point estimates of the element utilities from the full model, Model 1, and the simpler model, Model 2a, were not dissimilar: (differences ranged from 0 to 0.02, median 0.01). The simpler model was chosen since it required fewer parameter estimates, and consequently generated smaller standard errors. Model 2b was then employed as a means of treating the individual estimates of element utilities on equal terms, which led to appropriate values of standard error. The estimates of each of the utilities of the lowest elements, (A1, B1, C1, D1, and E1) were then translated to zero. Finally, multiplicative scaling set the estimate of the utility of the highest profile (A4, B3, C4, D3, E4) to 1000 to avoid fractional values while keeping the precision of the estimates to 2 decimal places, and the standard errors were adjusted accordingly. The result is a score on a scale with the zero-point representing 'minimal importance' and with an upper limit, 1000, representing a post-hoc choice of unit.

The rescaled utility estimates from model 2b are shown in Table 4. The two dimensions with the widest range of utility (lowest level to highest level) were *patient burden* and *intervention effect* while the dimension with the narrowest range of utility was *equity*.

The end-result is a method for calculating a score for each profile. The estimate of utility of a research question can be calculated by summing the estimates of utilities of its elements and the corresponding standard error can then be obtained using the estimate of the correlation matrix shown in the Table 5. For example, a research question that was evaluated as being important to all stakeholders (A4), for a disorder with mild symptoms (B1), was common (C3), studied treatment was potentially curative (D3) but did not address equity considerations (E1) would score 582 (SE 13.46).

## Testing the validity of ANZMUSC-RQIT using derived utility estimates

The mean ANZMUSC-RQIT score using utility estimates from the preceding study, in publications from high impact journals was 428 (SEM 13) compared to 341 (SEM 11) in publications from low impact journals ($p < 0.001$). The relationship between tertiles of ANZMUSC-RQIT scores and journal impact factor is shown in Table 6. Trials with higher scores were much more likely to be published in a high impact factor journal (OR 3.09 for tertile 2 and 6.78 for tertile 3, $p < 0.001$). The ANZMUSC-RQIT with stakeholder-derived utility scores discriminated much better between articles published in high and low impact journals than did RQIT using unweighted scoring (OR 0.88 for tertile 3).

The ANZMUSC-RQIT score was also significantly associated with log transformed citation count (standardized regression coefficient 0.33, $p < 0.001$), whereas there was no significant association between log transformed citation count and RQIT scores (standardized regression coefficient 0.07, $p = 0.32$).

## Discussion

There is a need for a transparent, fair and usable approach to identifying the most important research questions. We have presented a novel instrument that aims to provide a means of

**Table 4. Research Question Importance Tool (RQIT-ANZMUSC) utility values from the modified best-worst scaling survey.**

| Dimension | Element | *Utility estimate (SE) | |
|---|---|---|---|
| A. Extent to which the question is important to patients and other health decision-makers | A1 Not shown to be important to either patients or decision-makers | 0 | (6.8) |
| | A2 Shown to be important to health decision-makers but not patients | 17 | (5.5) |
| | A3 Shown to be important to patients but not decision-makers | 92 | (5.3) |
| | A4 Shown to be important to both patients and decision-makers | 198 | (7.3) |
| B. That it addresses an area of high patient burden | B1 Mild symptoms and little or no associated disability | 0 | (8.1) |
| | B2 Moderate symptoms and some disability | 150 | (6.6) |
| | B3 Significantly disabling, associated with mortality risk or no effective treatments available | 250 | (7.7) |
| C. That it addresses an area of high social burden | C1 Condition is rare (prevalence <0.1%) | 0 | (7.9) |
| | C2 Condition is somewhat common (0.1–1%) | 28 | (5.4) |
| | C3 Condition is common (1–10%) | 119 | (5.4) |
| | C4 Condition is highly prevalent (>10%) | 173 | (8.0) |
| D. Potential reduction in patient and/or social burden due to (clinical or implementation) intervention | D1 Symptomatic treatment only | 0 | (8.1) |
| | D2 Potential for intervention to treat both symptoms and underlying disease pathology | 166 | (6.4) |
| | D3 Potential for cure or fundamental alteration of disease course | 265 | (7.8) |
| E. Extent to which the question addresses health equity | E1 No information | 0 | (7.7) |
| | E2 Not relevant | 45 | (5.4) |
| | E3 Somewhat (may have some application to reduce health disparity) | 67 | (5.0) |
| | E4 Reducing health disparity is the focus | 114 | (8.3) |

* Re-scaled to make the lowest category in each dimension equal to 0 and the score of the highest profile (A4, B3, C4, D3, E4) equal to 1000.

ranking research questions in the area of MSK health intervention clinical trials. The instrument (ANZMUSC-RQIT), shown in Table 4, is based on a literature review and iterative stakeholder input about what constitutes important research, is sufficiently reliable when applied by consensus committee evaluation and is scored using utility values derived from the preferences expressed by a range of stakeholders, including consumers and clinicians. Furthermore, the scores from the tool were shown to be associated with higher quality clinical trial publications, as judged by journal Impact Factor and citation count.

This instrument could be used to easily assign a 'research question importance' score to a research proposal in contexts such as grant assessment panels or the evaluation of proposals for endorsement or implementation by clinical trial networks. The content of the instrument may also help researchers focus on formulating questions that have the greatest significance.

This work is linked to a broader approach undertaken by the Australian Clinical Trials Alliance (ACTA), of which ANZMUSC is a member. ACTA has also developed a scoring tool for assessing the merit of a research proposal using somewhat similar methodology to the work described here (manuscript in preparation) [24]. However, a key difference is that the ACTA

**Table 5. Correlation matrix for estimates of element utility from best-worst scaling exercise.** The entry for elements $e$ and $e'$ is $\rho_{e,e'}$*.

|  | A1 | A2 | A3 | A4 | B1 | B2 | B3 | C1 | C2 | C3 | C4 | D1 | D2 | D3 | E1 | E2 | E3 | E4 |
|---|---|---|---|---|---|---|---|---|---|---|---|---|---|---|---|---|---|---|
| A1 | 1.000 | -0.073 | -0.342 | -0.620 | 0.487 | -0.195 | -0.345 | 0.555 | 0.102 | -0.093 | -0.554 | 0.391 | -0.116 | -0.311 | 0.539 | 0.261 | -0.033 | -0.653 |
| A2 | -0.073 | 1.000 | -0.351 | -0.429 | 0.155 | -0.089 | -0.086 | 0.227 | -0.048 | -0.035 | -0.168 | 0.170 | -0.131 | -0.068 | 0.198 | -0.076 | -0.13 | -0.058 |
| A3 | -0.342 | -0.351 | 1.000 | -0.147 | -0.104 | 0.070 | 0.049 | -0.108 | -0.053 | -0.029 | 0.162 | -0.089 | 0.044 | 0.056 | -0.119 | -0.119 | 0.043 | 0.162 |
| A4 | -0.620 | -0.429 | -0.147 | 1.000 | -0.489 | 0.195 | 0.347 | -0.604 | -0.02 | 0.133 | 0.520 | -0.423 | 0.173 | 0.298 | -0.560 | -0.098 | 0.097 | 0.528 |
| B1 | 0.487 | 0.155 | -0.104 | -0.489 | 1.000 | -0.466 | -0.651 | 0.453 | 0.039 | -0.074 | -0.424 | 0.414 | -0.082 | -0.363 | 0.460 | 0.151 | -0.061 | -0.490 |
| B2 | -0.195 | -0.089 | 0.070 | 0.195 | -0.466 | 1.000 | -0.368 | -0.136 | 0.007 | -0.008 | 0.134 | -0.057 | -0.122 | 0.160 | -0.136 | -0.031 | -0.047 | 0.175 |
| B3 | -0.345 | -0.086 | 0.049 | 0.347 | -0.651 | -0.368 | 1.000 | -0.36 | -0.047 | 0.084 | 0.331 | -0.386 | 0.190 | 0.244 | -0.366 | -0.131 | 0.104 | 0.364 |
| C1 | 0.555 | 0.227 | -0.108 | -0.604 | 0.453 | -0.136 | -0.360 | 1.000 | -0.168 | -0.269 | -0.693 | 0.460 | -0.079 | -0.414 | 0.556 | 0.272 | -0.117 | -0.624 |
| C2 | 0.102 | -0.048 | -0.053 | -0.020 | 0.039 | 0.007 | -0.047 | -0.168 | 1.000 | -0.338 | -0.279 | 0.034 | -0.097 | 0.045 | 0.098 | -0.107 | 0.107 | -0.087 |
| C3 | -0.093 | -0.035 | -0.029 | 0.133 | -0.074 | -0.008 | 0.084 | -0.269 | -0.338 | 1.000 | -0.186 | -0.026 | -0.008 | 0.034 | -0.038 | -0.082 | -0.077 | 0.135 |
| C4 | -0.554 | -0.168 | 0.162 | 0.520 | -0.424 | 0.134 | 0.331 | -0.693 | -0.279 | -0.186 | 1.000 | -0.461 | 0.149 | 0.356 | -0.590 | -0.140 | 0.095 | 0.584 |
| D1 | 0.391 | 0.170 | -0.089 | -0.423 | 0.414 | -0.057 | -0.386 | 0.46 | 0.034 | -0.026 | -0.461 | 1.000 | -0.446 | -0.671 | 0.434 | 0.203 | -0.119 | -0.465 |
| D2 | -0.116 | -0.131 | 0.044 | 0.173 | -0.082 | -0.122 | 0.190 | -0.079 | -0.097 | -0.008 | 0.149 | -0.446 | 1.000 | -0.364 | -0.125 | -0.039 | 0.004 | 0.139 |
| D3 | -0.311 | -0.068 | 0.056 | 0.298 | -0.363 | 0.160 | 0.244 | -0.414 | 0.045 | 0.034 | 0.356 | -0.671 | -0.364 | 1.000 | -0.348 | -0.178 | 0.120 | 0.368 |
| E1 | 0.539 | 0.198 | -0.119 | -0.560 | 0.460 | -0.136 | -0.366 | 0.556 | 0.098 | -0.038 | -0.590 | 0.434 | -0.125 | -0.348 | 1.000 | -0.079 | -0.321 | -0.690 |
| E2 | 0.261 | -0.076 | -0.119 | -0.098 | 0.151 | -0.031 | -0.131 | 0.272 | -0.107 | -0.082 | -0.140 | 0.203 | -0.039 | -0.178 | -0.079 | 1.000 | -0.253 | -0.420 |
| E3 | -0.033 | -0.130 | 0.043 | 0.097 | -0.061 | -0.047 | 0.104 | -0.117 | 0.107 | -0.077 | 0.095 | -0.119 | 0.004 | 0.120 | -0.321 | -0.253 | 1.000 | -0.139 |
| E4 | -0.653 | -0.058 | 0.162 | 0.528 | -0.490 | 0.175 | 0.364 | -0.624 | -0.087 | 0.135 | 0.584 | -0.465 | 0.139 | 0.368 | -0.690 | -0.420 | -0.139 | 1.000 |

* Each entry is an estimate, $\rho_{e,e'}$ of a correlation coefficient. Calculate the SE for a profile score as $\sqrt{\sum_e \sum_{e'} \rho_{e,e'} \sigma_e \sigma_{e'}}$ summing over the 5 x 5 = 25 pairs of elements.

tool explicitly considers the overall merit of the research proposal including its appropriateness, feasibility, significance and relevance; whereas RQIT considers only the significance and relevance aspect of the proposal–what does society gain and why do the research in the first place? The ACTA tool encompasses additional domains of interest such as whether the research design is appropriate to its objective, and whether it is feasible for the question to be answered given constraints such as researcher expertise and available participants. Our focus on the importance of a research question may not be sufficient to determine whether a project *could* be done, but it does help address whether it *should* be done. We contend that the *appropriateness* and *feasibility* domains of the ACTA research prioritisation tool represent technical-methodological and resourcing issues. Furthermore, only research questions that are sufficiently important should receive the kind of investment needed to understand the technical and resource requirements required to answer the question.

There are some limitations to this work. Although consumers were involved in every step of data collection to construct the framework and to determine the scoring system, there were

**Table 6. Relationship between RQIT scores and journal impact factor group (low or high).** The odds ratios are obtained by logistic regression for tertiles of scores.

| Tertile | RQIT* | | ANZMUSC-RQIT† | |
|---|---|---|---|---|
|  | OR (95%CI) | P-value | OR (95%CI) | P-value |
| 1 | REF | | REF | |
| 2 | 0.38 (0.18 to 0.82) | 0.01 | 3.09 (1.49 to 6.42) | 0.003 |
| 3 | 0.88 (0.45 to 1.72) | 0.71 | 6.78 (3.17 to 14.50) | <0.001 |

* Logistic regression model chi-square 7.66 (2 df, p = 0.02), Nagelkere $R^2$, 0.05

† Logistic regression model chi-square 27.4 (2 df, p<0.001),Nagelkerke $R^2$ 0.17

relatively fewer consumers in comparison to other stakeholders (ranging from 9% of participants in the initial delphi survey to 18% of participants in the consensus workshop). Similarly, there were few funders of research or policy makers, who may have different perspectives on the relative utility of each dimension of RQIT. We did not attempt to determine whether there were systematic differences between stakeholder groups because of the relatively small size of the participant cohort. Nevertheless, in future research it will be of interest to test the element utilities in larger cohorts that might permit analysis of population segments.

The indices of high-quality research that were chosen (journal impact factor and citation count) could legitimately be open to challenge. Ideally the validity of the tool would be tested using better indicators of research impact, particularly influences on clinical practice or citations in clinical guidelines. This is more difficult to measure, and bibliometric indices were chosen mainly on pragmatic grounds. We did however show an association between risk of bias and journal impact factor, providing at least some evidence that more methodologically sound studies are published in higher impact factor journals. Further validity evaluation should be undertaken by assessment of how the tool works prospectively in practice and comparison of scores with citations in subsequent clinical guidelines.

Although the focus of the work concerned MSK health, the content of the RQIT dimension framework is likely to be relevant to other areas of health. It may be useful to test the tool beyond MSK health and particularly to compare the utilities from stakeholders in other areas of health. For now, we can recommend the use of the ANZMUSC-RQIT for evaluation of the importance of research questions that aim to test interventions that improve MSK health in randomised controlled trials.

## Supporting information

**S1 Appendix. Novel best-worst scaling technique.**
(PDF)

**S2 Appendix. The hypothetical research questions used to evaluation the reliability of RQIT.**
(PDF)

**S1 Table. Association of Cochrane Risk of Bias dimension categories with journal impact factor group.**
(PDF)

## Acknowledgments

We gratefully acknowledge the help of Emeritus Professor Jane Latimer, Helen Ramsay (Executive Officer of ANZMUSC) and Tania Pocock (Health Research Council of New Zealand), Dr Chris Dalton (National Medical Officer BUPA Australia) for participation in the consensus workshop. Sadly, Philip Robinson tragically died during submission of the manuscript following a short illness.

## Author Contributions

**Conceptualization:** William J. Taylor, Ian A. Harris, Samuel L. Whittle, Bethan Richards, Sally Green, Rana S. Hinman, Chris G. Maher, Paul Glasziou, Rachelle Buchbinder.

**Data curation:** William J. Taylor.

**Formal analysis:** William J. Taylor, Robin Willink.

**Funding acquisition:** Ian A. Harris, Rana S. Hinman, Chris G. Maher, Rachelle Buchbinder.

**Investigation:** William J. Taylor, Denise A. O'Connor, Vinay Patel, Allison Bourne, Ian A. Harris, Samuel L. Whittle, Bethan Richards, Ornella Clavisi, Sally Green, Rana S. Hinman, Chris G. Maher, Ainslie Cahill, Annie McPherson, Charlotte Hewson, Suzie E. May, Bruce Walker, Philip C. Robinson, Davina Ghersi, Jane Fitzpatrick, Tania Winzenberg, Kieran Fallon, Paul Glasziou, Laurent Billot, Rachelle Buchbinder.

**Methodology:** William J. Taylor, Robin Willink, Laurent Billot.

**Project administration:** Allison Bourne, Rachelle Buchbinder.

**Resources:** Philip C. Robinson.

**Supervision:** Ian A. Harris, Rachelle Buchbinder.

**Writing – original draft:** William J. Taylor.

**Writing – review & editing:** Robin Willink, Denise A. O'Connor, Vinay Patel, Allison Bourne, Ian A. Harris, Samuel L. Whittle, Bethan Richards, Ornella Clavisi, Sally Green, Rana S. Hinman, Chris G. Maher, Ainslie Cahill, Annie McPherson, Charlotte Hewson, Suzie E. May, Bruce Walker, Philip C. Robinson, Davina Ghersi, Jane Fitzpatrick, Tania Winzenberg, Kieran Fallon, Paul Glasziou, Laurent Billot, Rachelle Buchbinder.

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
