## [Decision Letter · Decision Letter 0]

4 Oct 2022

PONE-D-22-23846Development and preliminary validation of the Australia & New Zealand Musculoskeletal (ANZMUSC) Clinical Trials Network Research Question Importance Tool (RQITv2.0)PLOS ONE

Dear Dr. Taylor,

Thank you for submitting your manuscript to PLOS ONE. After careful consideration, we feel that it has merit but does not fully meet PLOS ONE’s publication criteria as it currently stands. Therefore, we invite you to submit a revised version of the manuscript that addresses the points raised during the review process.

We look forward to receiving your revised manuscript.

Kind regards,

Charlotte Beaudart

Academic Editor

PLOS ONE

Journal Requirements:

"This study was partly funded by the NHMRC (Centre of Research Excellence: Australia & New Zealand Musculoskeletal (ANZMUSC) Clinical Trials Network; Chief Investigator Rachelle Buchbinder). The funders had no role in study design, data collection and analysis, decision to publish, or preparation of the manuscript."

"We gratefully acknowledge the help of Emeritus Professor Jane Latimer, Helen Ramsay (Executive Officer of ANZMUSC) and Tania Pocock (Health Research Council of New Zealand), Dr Chris Dalton (National Medical Officer BUPA Australia) for participation in the consensus workshop. RB is supported by an Australian National Health and Medical Research Council (NHMRC) Investigator Fellowship (APP1194483). RSH is supported by a NHMRC Senior Research Fellowship (#1154217). ANZMUSC is supported by a Centre of Research Excellence Grant by NHMRC (APP1134856)."

"This study was partly funded by the NHMRC (Centre of Research Excellence: Australia & New Zealand Musculoskeletal (ANZMUSC) Clinical Trials Network; Chief Investigator Rachelle Buchbinder). The funders had no role in study design, data collection and analysis, decision to publish, or preparation of the manuscript."

6. One of the noted authors is a group or consortium ANZMUSC Clinical Trial Network. In addition to naming the author group, please list the individual authors and affiliations within this group in the acknowledgments section of your manuscript. Please also indicate clearly a lead author for this group along with a contact email address.

Reviewers' comments:

Reviewer's Responses to Questions

**Comments to the Author**

1. Is the manuscript technically sound, and do the data support the conclusions?

Reviewer #1: Yes

Reviewer #2: Yes

2. Has the statistical analysis been performed appropriately and rigorously? 

Reviewer #1: Yes

Reviewer #2: Yes

3. Have the authors made all data underlying the findings in their manuscript fully available?

Reviewer #1: No

Reviewer #2: Yes

4. Is the manuscript presented in an intelligible fashion and written in standard English?

Reviewer #1: Yes

Reviewer #2: Yes

5. Review Comments to the Author

Reviewer #1: The topic is very important and some of the tools used by the authors are either inoovative or scientifically robust and the authors are to be commended for this. However, there are a number of problems that limit the validity of the research or that limit the understanding of the paper:

- The title is not very clear. Please update it so that it really reflects the work of the authors.

- The developers of this tool are all from Australia or New Zealand. If the goal is to provide a tool specifically for Australia and New Zealand, I wonder if this merits an international publication.

- The inclusion of "version 2.0" in the name of the tool is somewhat misleading as there is no real version 1. The "new" version is a simple and logical evolution of the tool during its development.

- All (or most) of the experts are in the field of musculoskeletal disorders and therefore the tool cannot be considered a generic tool. This should be reflected in the name of the tool so that it will not be used in a bad way.

- The main problem with this tool is how the initial validation is done. IF and citation counts are very poor and totally arbitrary proxies for research quality. This is acknowledged by the authors, but it does not solve the problem. At least, some sentitivity analyses with the use of other journals can be made (and not anly to 5 top and down journals).

- Surprisingly, contrary to current practice, patients were not considered in the development of the tool and the potential consequence of this is that the patient perspective is not considered a potential attribute.

- Much more information needs to be provided on the 14 hypothetical research questions. If these research questions are written in a way that makes scoring easy, the bias is too great. I strongly recommend using the previously submitted applications.

- The authors state that this is a preliminary validation but they do not clearly explain what their next steps are and why this was not done for this article.

- For full transparency, it is important to include in the paper the full final version of the tool or to explain how to access it and to explain very clearly how to score it.

Reviewer #2: The subject of the article: Development and preliminary validation of the Australia & New Zealand Musculoskeletal (ANZMUSC) Clinical Trials Network Research Question Importance Tool (RQITv2.0).

After reading the article, the reviewer found it to be an original and innovative manuscript. The manuscript is not characterized by significant formal deficiencies and factors that interfere with interpretation. The style and language are booth good, the structure is understandable, and the authors have devoted plenty of time to providing the necessary background knowledge.

The abstract summarizes the essence of the article well.

In the introduction, the reviewer is slightly missing additional data in a global context, or in comparison with other countries. According to the reviewer, it should be considered to correct this.

The method part has been precisely defined. The methods are well described, authentic and can be followed, the results presented in the tables are understandable and convincing. The description of the results and the use of tables are appropriate.

The discussion is satisfactory both in terms of content and scope.

In summary, all of these give the manuscript its significance, while I detailed above that the authors solved this excellently. the reviewer recommends the article for publication, which, in the reviewer's opinion, contains valuable data and information.

6. PLOS authors have the option to publish the peer review history of their article (what does this mean?). If published, this will include your full peer review and any attached files.

Reviewer #1: No

Reviewer #2: No

---

## [Author Response · Author response to Decision Letter 0]

16 Jan 2023

Re: PONE-D-22-23846

Development and preliminary validation of the Australia & New Zealand Musculoskeletal (ANZMUSC) Clinical Trials Network Research Question Importance Tool (RQITv2.0)

Charlotte Beaudart

Academic Editor

PLoS One 

Dear Dr Beaudart

Many thanks for allowing us the opportunity of re-submission. I have tabulated our response to the reviewers’ and editor’s comments. Any references to the text refers to the re-submitted version of the manuscript.

The funding statement should state: 

This study was funded by the Australian National Health and Medical Research Council (NHMRC) Australia & New Zealand Musculoskeletal (ANZMUSC) Clinical Trials Network Centre of Research Excellence (APP1134856). RB is supported by an NHMRC Investigator Fellowship (APP1194483). RSH is supported by an NHMRC Senior Research Fellowship (APP1154217). The funders had no role in study design, data collection and analysis, decision to publish, or preparation of the manuscript.

Editorial comments 

1. File naming and format requirements; Response - We have re-named files and formatted the manuscript to conform to style requirements Level 1 headings bold and 18pt font; level 2 headings bold and 16pt font, Figures and Table citations have been changed. Supplementary files have been created.

2. Mismatch of ‘funding information’ and ‘financial disclosure’ statements; Response - Correct grants numbers are now provided 

3. Funding statement; Response - Amended funding statement to include all funding and sources of support, in cover letter above. 

4. Acknowledgements section; Response - Funding information moved to the Funding Statement as suggested. 

5. Data availability; Response - We have added data files to the repository to permit a minimal data-set. 

6. Group authorship; Response - We wish to withdraw the group author as all authors are already individually included. Author list is amended.

Reviewer 1 comments 

1. Unclear title; Response - We have changed the title to emphasize the context of the assessment tool. Title is now changed from: “Development and preliminary validation of the Australia & New Zealand Musculoskeletal (ANZMUSC) Clinical Trials Network Research Question Importance Tool (RQITv2.0)” to: “Which clinical research questions are the most important? Development and preliminary validation of the Australia & New Zealand Musculoskeletal (ANZMUSC) Clinical Trials Network Research Question Importance Tool”

2. The developers of this tool are all from Australia or New Zealand. If the goal is to provide a tool specifically for Australia and New Zealand, I wonder if this merits an international publication; Response - The goal is to provide a tool applicable in any geographic location. There are no aspects of the work that relate to local factors. No changes made.

3. Naming of the tool version; Response - We agree that this is potentially misleading, but we needed a means of referring to different versions of the tool as it developed during the study. We have removed 2.0 from the name of the final version of the tool and have used a different naming system within the manuscript to facilitate accurate understanding.

4. MSK specific tool; Response - We agree and have referred to the tool as Australia and New Zealand Musculoskeletal Clinical Trials Network Research Question Importance Tool (ANZMUSC-RQIT) to reflect this. ANZMUSC appended to final tool name in title and throughout manuscript.

5. Validation; Response - As noted in the limitations section of the manuscript, we agree that validation is challenging and bibliographic indices are far from ideal. However, the suggested sensitivity analysis will only compound this problem by continuing to employ bibliographic indices that will have even less certain meaning. A key step in our approach was to confirm a relationship exists between clinical trial quality as assessed by the Cochrane RoB tool and journal impact factor (see page 15, line 316-318 in methods and page 26 lines 462-464 in results (and also Table S1). We believe this is a better way to provide evidence to support our approach. No changes made.

6. Patient involvement; Response - We are surprised by the reviewer’s comment and wonder if our use of the term ‘consumer’ was somehow misunderstood. We have therefore clarified in the manuscript that consumers included patients and described their involvement in more detail. Additional text ln195-201 on pg 11: “Consumers (health-service users) were involved at every stage of the project, including generation of possible attributes of research importance (9% of participants), consensus development of the initial RQIT framework (18% of participants), weighting of specific elements of the framework using a novel best-work scaling method (12% of participants) and final authorship of this manuscript. Consumers were identified from membership of ANZMUSC and all had lived experience of musculoskeletal disorders.”

7. Hypothetical research questions; Response - These are now provided as a supplementary appendix. We do not believe it is necessary to repeat the reliability analysis. It is sufficiently clear that scoring by individuals is not reliable and only scoring by committee consensus is adequately reliable. New supplementary appendix.

8. Next steps for validation; Response - This is a fair comment. The main next step for assessing the tool will come from actual use in practice. We thought it was important that the tool be made available for other groups to use by having the work published. We argue that it is sufficiently ready for practical use but that further evaluation will only come from experience in practical settings. That could not be part of the current development phase which is described in this manuscript. Additional text in the Discussion section page 34, lines: 590-592 “Further validity evaluation should be undertaken by assessment of how the tool works prospectively in practice and comparison of scores with citations in subsequent clinical guidelines.”

9. Include the full final version and scoring instructions; Response - This was already provided as Table 4 and a worked example was provided. No changes made.

Reviewer 2 comments 

1. Global context or comparison with other countries; Response - We appreciate that Australian and New Zealand data was used to illustrate MSK burden of disease, but do note that the introduction refers to the Global Burden of Disease project. We have added information from the US. New text added to introduction: “In the United States, funding for the National Institute of Arthritis and Musculoskeletal and Skin Diseases (NIAMS) have never been more than 2% of the National Institutes of Health annual disbursement (7)”. 

2. Positive comments; Response - We appreciate the reviewer’s favourable opinion on the merits of the manuscript.

---

## [Decision Letter · Decision Letter 1]

20 Jan 2023

Which clinical research questions are the most important? Development and preliminary validation of the Australia & New Zealand Musculoskeletal (ANZMUSC) Clinical Trials Network Research Question Importance Tool (ANZMUSC-RQIT)

PONE-D-22-23846R1

Dear Dr. Taylor,

We’re pleased to inform you that your manuscript has been judged scientifically suitable for publication and will be formally accepted for publication once it meets all outstanding technical requirements.

Kind regards,

Charlotte Beaudart

Academic Editor

PLOS ONE

Additional Editor Comments (optional):

Reviewers' comments:

Reviewer's Responses to Questions

**Comments to the Author**

1. If the authors have adequately addressed your comments raised in a previous round of review and you feel that this manuscript is now acceptable for publication, you may indicate that here to bypass the “Comments to the Author” section, enter your conflict of interest statement in the “Confidential to Editor” section, and submit your "Accept" recommendation.

Reviewer #1: All comments have been addressed

2. Is the manuscript technically sound, and do the data support the conclusions?

Reviewer #1: Yes

3. Has the statistical analysis been performed appropriately and rigorously? 

Reviewer #1: Yes

4. Have the authors made all data underlying the findings in their manuscript fully available?

Reviewer #1: Yes

5. Is the manuscript presented in an intelligible fashion and written in standard English?

Reviewer #1: Yes

6. Review Comments to the Author

Reviewer #1: ok

7. PLOS authors have the option to publish the peer review history of their article (what does this mean?). If published, this will include your full peer review and any attached files.

Reviewer #1: No

---

## [Editor Report · Acceptance letter]

26 Jan 2023

PONE-D-22-23846R1 

Which clinical research questions are the most important? Development and preliminary validation of the Australia & New Zealand Musculoskeletal (ANZMUSC) Clinical Trials Network Research Question Importance Tool (ANZMUSC-RQIT) 

Dear Dr. Taylor:

I'm pleased to inform you that your manuscript has been deemed suitable for publication in PLOS ONE. Congratulations! Your manuscript is now with our production department. 

Kind regards, 

on behalf of

Dr. Charlotte Beaudart 

Academic Editor

PLOS ONE